# DISSECTING LOCAL PROPERTIES OF ADVERSARIAL EXAMPLES

## ABSTRACT

Adversarial examples have attracted significant attention over the years, yet a sufficient understanding is in lack, especially when analyzing their performances in combination with adversarial training. In this paper, we revisit some properties of adversarial examples from both frequency and spatial perspectives: 1) the special high-frequency components of adversarial examples tend to mislead naturally-trained models while have little impact on adversarially-trained ones, and 2) adversarial examples show disorderly perturbations on naturally-trained models and locally-consistent (image shape related) perturbations on adversarially-trained ones. Motivated by these, we analyze the fragile tendency of models with the generated adversarial perturbations, and propose a connection with model vulnerability and local intermediate response. That is, a smaller local intermediate response comes along with better model adversarial robustness. To be specific, we demonstrate that: 1) DNNs are naturally fragile at least for large enough local response differences between adversarial/natural examples, 2) and smoother adversarially-trained models can alleviate local response differences with enhanced robustness.

## 1 INTRODUCTION

Despite deep neural networks (DNNs) perform well in many fields (He et al., 2016; Devlin et al., 2019), their counter-intuitive vulnerability attracts increasing attention, both for safety-critical applications (Sharif et al., 2016) and the black-box mechanism of DNNs (Fazlyab et al., 2019). DNNs have been found vulnerable to adversarial examples (Szegedy et al., 2014; Goodfellow et al., 2015), where small perturbations on the input can easily change the predictions of a well-trained DNN with high confidence. In computer vision, adversarial examples exhibit their destructiveness both in the digital world and the physical world (Kurakin et al., 2017).

Since then, how to alleviate the vulnerability of DNN so as to narrow the performance gap between adversarial/natural examples is another key issue. Existing methods including defensive distillation (Papernot et al., 2016) and pixel denoising (Liao et al., 2018) have shown their limitations due to follow-up attack strategies (Carlini & Wagner, 2017) or gradient masking (Athalye et al., 2018). Amongst them, adversarial training (Goodfellow et al., 2015; Madry et al., 2018) and its variants (Zhang et al., 2019; Wang et al., 2020b) indicate their reliable robustness and outperform. Moreover, as a data augmentation method, adversarial training currently seems to rely on additional data (Schmidt et al., 2018; Rebuffi et al., 2021) to further improve robustness, while is sensitive to some basic model hyper-parameters, e.g., weight decay (Pang et al., 2021). Apart from these, the effect of simply early stopping (Rice et al., 2020) even exceeds some promotion methods according to recent benchmarks (Croce & Hein, 2020; Chen & Gu, 2020). These studies arise our curiosity to further explore the relationship between adversarial examples and adversarial training, hoping to provide some new understanding.

Recalling that high-frequency components can be potentially linked to adversarial examples (Wang et al., 2020a; Yin et al., 2019; Harder et al., 2021), however, few explorations discuss the relationship between high-frequency components and the destructiveness of adversarial examples. In this paper, we first demonstrate that high-frequency components of adversarial examples tend to mislead the standard DNNs, yet little impact on the adversarially robust models. We further show that adversarial examples statistically have more high-frequency components than natural ones, indicating relatively drastic local changes among pixels of adversarial examples. Since adversarial examples

exhibit more semantically meaningful on robust models (Tsipras et al., 2019), we further notice that adversarial examples show locally-consistent perturbations related to image shapes on adversarially-trained models, in contrast to disorderly perturbations on standard models. Both explorations on the frequency and spatial domain emphasize local properties of adversarial examples, and motivated by the local receptive field of the convolution kernels, we propose a locally intermediate response perspective to rethink the vulnerability of DNNs. Different from the existing global activation perspective (Bai et al., 2021; Xu et al., 2019), our local perspective reflects the joint effect of local features and the intermediate layers of the model. Based on the local perspective, we emphasize that large enough local response differences make it difficult for the network to treat an image and its potentially adversarial examples as one category, and demonstrate DNN models are naturally fragile at least attributed to it. Motivated by adversarially-trained models tend to have 'smooth' kernels (Wang et al., 2020a), we simply use the smoother kernels to alleviate local response differences on adversarially-trained models, which in turn affects the model robustness and reduces the robust overfitting (Rice et al., 2020). To a certain extent, this explains why weight decay effectively affects model robustness (Pang et al., 2021).

Our main contributions are summarized as follows:

- We first reveal some properties of adversarial examples in the frequency and spatial domain: 1) the high-frequency components of adversarial examples tend to mislead naturally-trained DNNs, yet have little impact on adversarially-trained models, and 2) adversarial examples have locally-consistent perturbations on adversarially-trained models, compared with disorderly local perturbations on naturally-trained models.

- Then we introduce local response and emphasize its importance in the model adversarial robustness. That is, naturally-trained DNNs are often fragile, at least for non-ignorable local response differences through the same layer between potentially adversarial examples and natural ones. In contrast, adversarially-trained models effectively alleviate the local response differences. And the smoother adversarially-trained models show better adversarial robustness as they can reduce local response differences.

- Finally we empirically study local response with generated adversarial examples. We further show that, compared with failed attacks, adversarial examples (successful attacks) statistically show larger local response differences with natural examples. Moreover, compared with adversarial examples generated by the model itself, those transferred by other models show markedly smaller local response differences.

## 2  RELATED WORK

**Understandings of model vulnerability.** Since the discovery of adversarial examples (Szegedy et al., 2014), a number of understandings on model vulnerability has been developed. For instance, linear property of DNNs (Goodfellow et al., 2015), submanifold (Tanay & Griffin, 2016) and geometry of the manifold (Gilmer et al., 2018) were considered from the high-dimensional perspective; the computing of Lipschitz constant (Szegedy et al., 2014; Fazlyab et al., 2019) and lower/upper bounds (Fawzi et al., 2018; Weng et al., 2018) were considered from the definition of model robustness; non-robust features (Ilyas et al., 2019), high-frequency components (Wang et al., 2020a; Yin et al., 2019), high-rank features (Jere et al., 2020) and high-order interactions among pixels (Ren et al., 2021) were explored adversarial examples from the different perspectives on images, which imply our local perspective; feature denosing (Xie et al., 2019), robust pruning (Madaan & Ju Hwang., 2020) and activation suppressing (Bai et al., 2021) were focused on global intermediate activations of models. On the other hand, taken adversarial training into consideration, Tsipras et al. (2019), Schmidt et al. (2018) and Zhang et al. (2019) explored the trade-off between robustness and accuracy; Wang et al. (2020a) found adversarially-trained models tend to show smooth kernels; Tsipras et al. (2019) and Zhang & Zhu (2018) argued adversarially robust models learned more shape-biased representations.

Different from these studies, we characterize the adversarial examples from both frequency and spatial domain to emphasize local properties of adversarial examples, and propose a locally intermediate response to rethink the vulnerability of DNNs.

**Adversarial training.** Adversarial training can be seen as a min-max optimization problem:

$$\min_{\theta} \frac{1}{n} \sum_{i=1}^{n} \max_{x_i' \in B(x)} L(f_\theta(x_i'), y_i)$$

where $f$ denotes a DNN model with parameters $\theta$, and $(x_i, y_i)$ denotes a pair of a natural example $x_i$ and its ground-truth label $y_i$. Given a classification loss $L$, the inner maximization problem can be regarded as searching for suitable perturbations in boundary $B$ to maximize loss, while the outer minimization problem is to optimize model parameters on adversarial examples $\{x_i'\}_{i=1}^{n}$ generated from the inner maximization.

## 3 DIFFERENT PERTURBATIONS AFFECT THE VULNERABILITY OF THE MODEL

In this section, we first investigate adversarial examples in the frequency and spatial domain, and show the connections between models and their adversarial examples. Note that our threat model is a white-box model, and the fact that the adversarial example is only defined by the misleading result under the legal threat model, our findings suggest that the properties of the sampled examples can broadly reflect the fragile tendency of the model.

**Setup.** We generate $\ell_\infty$ bounded adversarial examples by PGD-10 (maximum perturbation $\epsilon = 8/255$ and step size $2/255$) with random start for the robust model (Madry et al., 2018). Specifically, we use ResNet-18 (He et al., 2016) as the backbone to train the standard and adversarially-trained models for 100 epochs on CIFAR-10 (Krizhevsky, 2009). Following (Pang et al., 2021), we use the SGD optimizer with momentum 0.9, weight decay $5 \times 10^{-4}$ and initial learning rate 0.1, with a three-stage learning rate divided by 10 at 50 and 75 epoch respectively. The robustness of both models is evaluated by PGD-20 (step size $1/255$).

### 3.1 THE DESTRUCTIVENESS OF HIGH-FREQUENCY COMPONENTS FROM ADVERSARIES

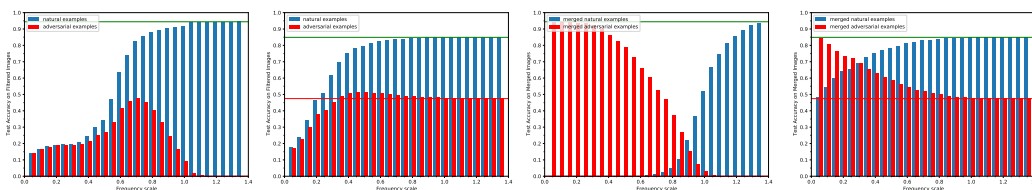

(a) filtered images (STD)  (b) filtered images (ADV)  (c) merged images (STD)  (d) merged images (ADV)

Figure 1: The destructiveness of high-frequency components from natural and adversarial examples on both standard (STD) and adversarially-trained (ADV) models. Shown above are well-trained models tested with (a)-(b) images through low-pass filter and (c)-(d) frequency-swapped images.

Inspired by (Wang et al., 2020a), we are naturally curious whether the adversarial examples cause considerable damage to the model mainly because of their high-frequency components. To answer this question, Figure 1 illustrates the trend of model performance and robustness on the test set with the high-frequency components increased (the increase of the filtering scale denotes that more high-frequency components are added to the filtered images). Figure 1(a) shows that, for standard models, the high-frequency components of natural examples promote classification and reach the model performance (green line); on the contrary, the performance of the filtered adversarial examples first rises to get the highest accuracy 47.5%, and then drops rapidly to reach 0.0% (red line). Obviously, in the low-frequency range, the performance of natural and adversarial examples are quite close, *yet more high-frequency components widen the difference*. That is, the special high-frequency components caused by adversarial perturbations exhibit a clear destructive effect on standard models, and simply filter out them can effectively alleviate the destructiveness of adversaries even on standard models.

However, for robust models, we show that the prediction performance finally reaches robustness without a rapid drop in Figure 1(b). But surprisingly, we find that the performance of filtered adversarial examples in some range exceeds the final robustness 47.5% (red line), reaching a maximum of 51.2%. That is, although these high-frequency components do not exhibit a clear destructive effect, simply filtering out them has a positive impact on alleviating robust overfitting (Rice et al., 2020).

We then swap their high-frequency components between both examples controlled by a frequency threshold in Figure 1(c)-1(d). For merged natural examples with high-frequency components from adversaries, the increase of the frequency threshold controls the accuracy increasing from the model robustness (red line) to the model performance (green line), the opposite occurs on merged adversarial examples. These clearly illustrate the boost effect of the high-frequency components from natural examples and the destructive effect of the high-frequency components from adversarial examples.

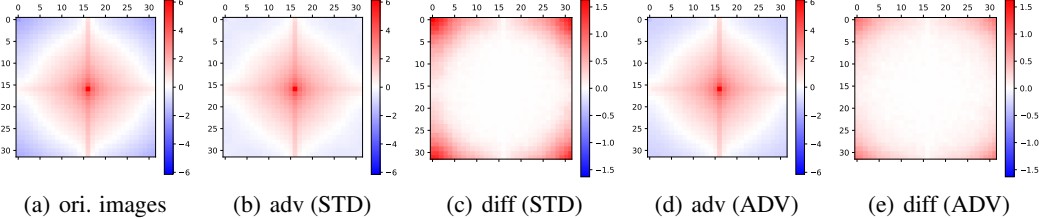

| (a) ori. images | (b) adv (STD) | (c) diff (STD) | (d) adv (ADV) | (e) diff (ADV) |

Figure 2: The average logarithmic amplitude spectrum of (a) 1000 three-channel images and (b) their adversarial examples generated by the standard (STD) model, where the corners represent high-freq range, and the colorbars represent the logarithmic amplitude $log(|\cdot|)$ (the redder the larger). And (c) denotes the difference between (b) and (a), $log(|adv|) - log(|nat|) = log(|adv|/|nat|)$, where the write color of (c) represents equivalent, and the red represents $|adv| > |nat|$. (d) and (e) are on the adversarially-trained (ADV) model, and (e) denotes the difference between (d) and (a).

To further illustrate, we find that statistically, the main difference in the frequency domain between natural and adversarial examples is concentrated in the high-frequency region in Figure 2. We visualize the logarithmic amplitude spectrum of both examples. Figure 2(a) and 2(b) show that, compare with natural examples', the high-frequency components of the adversaries are hard to ignore. Figure 2(c) further emphasizes that adversarial examples markedly show more high-frequency components, *indicating relatively drastic local changes among pixels*. This statistical difference explains the high detection rate of using Magnitude Fourier Spectrum (Harder et al., 2021) to detect adversarial examples. Furthermore, Figure 2(d) and 2(e) show that the high-frequency components of adversarial examples generated by robust models are less than those from standard models, yet still more than natural examples'. Besides, the analysis of filtering out low-frequency components in Figure 6 (Appendix A) also emphasizes our statement. That is, compared with natural examples', the special high-frequency components of adversarial examples show their serious misleading effects on standard models, yet are not enough to be fully responsible for the destructiveness to robust models.

## 3.2 DIFFERENT PERTURBATIONS AND THE FRAGILE TENDENCY OF THE MODEL

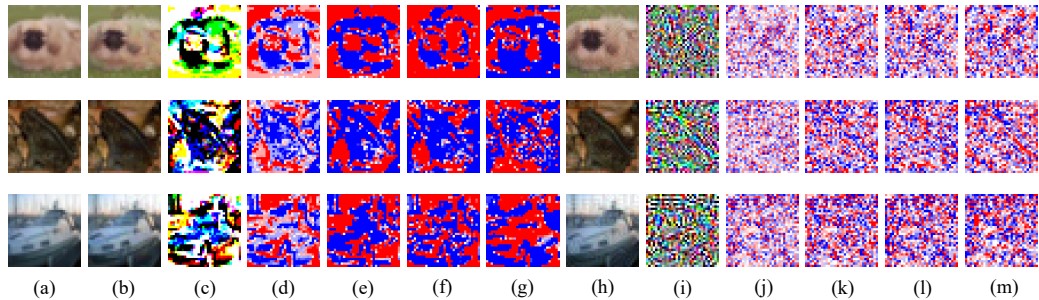

|  (a)  |  (b)  |  (c)  |  (d)  |  (e)  |  (f)  |  (g)  |  (h)  |  (i)  |  (j)  |  (k)  |  (l)  |  (m)  |

Figure 3: Visualisation of adversarial perturbations (PGD) generated by Left: adversarially-trained (ADV) model and Right: standard (STD) model. The columns from left to right are represented by (a)-(m) respectively: (a) natural example; (b) adversarial example and (c)-(g) its perturbations (overall, average of channels, and three channels) generated by ADV model; (h) adversarial example and (i)-(m) its perturbations generated by STD model. Shown above are all successfully attacked, the first row is attacked as a bird from a dog, the second a cat from a frog, the third a car from a ship.

To answer the different effects of high-frequency components, we directly explore the adversarial examples themselves, e.g., in the spatial domain. We first get well-trained models as above and visualize adversarial perturbations in Figure 3, including the overall perturbations scaled to [0,255], the average perturbations of channels, and perturbations of three channels.

As shown in Figure 3(Right), the perturbations generated by the standard models tend to be locally inconsistent and disordered. This observation is different from Figure 3(Left) that adversarial examples show locally-consistent perturbations related to image shapes on the adversarially-trained models, that is, perturbations tend to be locally co-increasing or co-decreasing on each channel. In more detail, the perturbation of each pixel on a single channel tend to reach the perturbation bound, i.e., +8/255 (the reddest), -8/255 (the bluest), which is counter-intuitive under the iterative attack of PGD-20 and naturally associated with the one-step attack FGSM in Figure 7 (Appendix B). In fact, both attacks show similar perturbations and similar model robustness[1], while the former produces more detailed perturbations. Besides, compared with failed attacks in Figure 8 (Appendix B), adversarial examples of successful attacks in Figure 3 show more misleading local perturbations, e.g., the perturbations in the first row are more like a bird (bird wings are added), and the third like a car.

**Perturbations in frequency vs. spatial domain.** Similar to the different destructive effects of high-frequency components, the perturbations in the spatial domain exhibit a more intuitive difference related to the models. Since more high-frequency components indicate images have relatively drastic local changes, we count that adversarial examples generated by adversarially-trained models show fewer high-frequency components mainly due to their locally-consistent perturbations. These perturbations with smooth local changes imply that simply filtering out high-frequency components has little promotion on robust models, while locally-disordered perturbations imply the effectiveness of filtering out high-frequency components on standard models.

**Perturbations vs. fragile trend of models.** For a given model, optimization-based attacks attempt to search for special perturbations to maximize the classification loss of adversarial examples. We note that for adversarially-trained models with smooth kernels, the perturbations that tend to maximize loss exhibit a locally-consistent tendency, and for standard models with non-smooth kernels, the perturbations tend to be locally-disordered to maximize loss. This implies a potential connection between the fragile trend of models and their potentially adversarial examples.

# 4 LOCAL RESPONSES: FURTHER IMPACT OF PERTURBATIONS ON MODELS

In this section, motivated by local properties of adversarial examples and the local receptive field of convolution kernels, we first introduce a locally intermediate response perspective to rethink the vulnerability of the model, and then empirically show the relationship between local responses and the vulnerability of the model.

Different from the existing idea of searching for adversarial examples, we consider *under what circumstances the potential examples, refer to all legal variants in boundary B regardless of whether destructive or not, would show their destructive effects on a well-trained model*, that is, the reason why some macro-similar potential examples present prediction results far away from their natural examples. Inspired by local properties of adversarial examples and kernels, we naturally consider whether the destructiveness of potential examples can be viewed from local responses, which reflects the combined effect of the local features and the model property.

Note that under ideal circumstances, given a well-trained model, if the local responses of any examples on a certain layer are completely consistent, then the final predictions of these examples are exactly the same. To relax the condition in reality, we hypothesize as follows (referred to as **Assumption 1**): *If macro-similar features through the same layer exhibit a sufficiently small difference in local responses, with sufficiently small differences in the subsequent layers, then the final responses of the network are relatively close;* otherwise, the large enough local differences make the network difficult to treat them as the same category. To illustrate this, we take the convolutional layer as an example.

## 4.1 LOCAL RESPONSES

Macroscopically, we denote a DNN model $f$ for classification, the $l$-th layer of activation feature maps $f^l$, and $f^0$ to represent the input features. Macro-similar image and its potential example pass through the $l$-th layer to get $f^l$ and $\hat{f}^l$ respectively. We also denote the $(l+1)$-th weight com-

---

[1]The adversarially-trained model (the best checkpoint) reaches 51.9% robustness on the PGD-20 attack and 57.4% robustness on the FGSM attack.

ponent $M^{l+1} \in \mathbb{R}^{H \times W \times K}$, where $H$, $W$, $K$ represent height, weight and number of kernels respectively. From the locally intermediate response perspective, we first get one of the convolution kernels $m^{l+1} \in \mathbb{R}^{H \times W}$, and then capture the $l$-th layer of local features centered at (i,j) position $x_{i,j}^l$ and $\hat{x}_{i,j}^l$ corresponding to its local receptive field. Formally, for the difference of local responses,

$$\Delta_{i,j}^{l+1} := \hat{x}_{i,j}^l \otimes m^{l+1} - x_{i,j}^l \otimes m^{l+1} = x_d{}^l \otimes m^{l+1}$$

where $\Delta_{i,j}^{l+1}$ denotes the (i,j)-th response difference of local features through the same kernel $m^{l+1}$, and $x_d{}^l := \hat{x}_{i,j}^l - x_{i,j}^l$ denotes the difference of local feature maps between its potential example and natural example. The second equation is based on the linear property of the convolution operation.

Ideally, if the differences are all equal to zeros at any positions in the $(l+1)$-th layer, that is, the intermediate layer has the same utility for both examples, then the potential example after the $(l+1)$-th layer shows exactly the same responses as the natural example's. However, in most cases the local differences are difficult to be exactly zeros everywhere, then based on Assumption 1, our aim is to make the absolute difference of local responses $|\Delta_{i,j}^{l+1}|$ as small as possible to approach the model's cognition of potential example and natural example.

The difference of local responses $\Delta_{i,j}^{l+1}$ composed of $x_d{}^l$ and $m^{l+1}$ can be further expressed as:

$$x_d{}^l \otimes m^{l+1} = \sum_{i=1}^{H} \sum_{j=1}^{W} c_{ij}^l m_{ij}^{l+1}$$

where $c_{ij}^l$ and $m_{ij}^{l+1}$ represent the (i,j)-th elements of $x_d{}^l$ and $m^{l+1}$ respectively, thus the absolute local difference $|\Delta_{i,j}^{l+1}|$ is affected jointly by both local features and kernel parameters.

Note that what we care about is under what circumstances are more likely to produce large enough response differences. Though the specific impact of $c_{ij}^l$ and $m_{ij}^{l+1}$ on $\Delta_{i,j}^{l+1}$ is quite complicated, we consider statistical circumstances as numerous convolution kernels and various local features are combined to affect the response differences in real networks. Statistically, the larger amplitude of $m^{l+1}$ with fixed $x_d{}^l$, or larger amplitude of $x_d{}^l$ with fixed $m^{l+1}$, tends to produce larger $|\Delta_{i,j}^{l+1}|$. Besides, if a kernel $m^{l+1}$ is relatively non-smooth, then non-smooth local features $x_d{}^l$, rather than smooth local features, are more likely to produce large enough local response differences.

## 4.2 FURTHER IMPACT OF LOCAL PERTURBATIONS ON MODELS

Recalling that local properties of adversarial examples in Section 3, we further explore how different local perturbations affect the models from the locally intermediate response perspective. To illustrate the effect of perturbations, we assume two convolution kernels with the same mean but different variances, one is disordered, and the other is smoother.

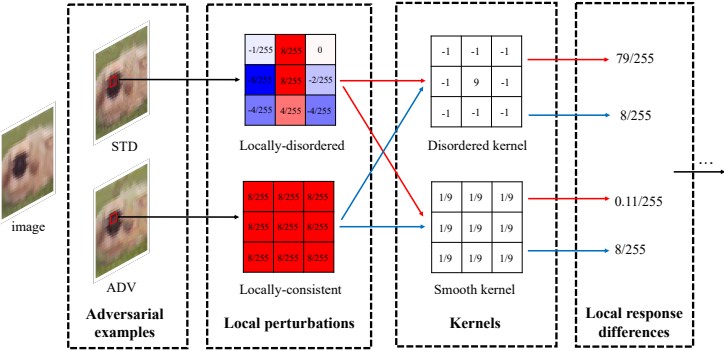

Figure 4: Further impact of local perturbations on different convolutional kernels. Shown above is adversarial perturbations with same amplitude (single channel) act on kernels with different smoothness and then get the local response differences of adversarial example and natural example.

**Local responses vs. locally-disordered perturbations.** Since adversarial perturbations generated by standard models tend to be locally-disordered, we discuss these perturbations affect the local responses. Locally-disordered perturbations indicate relatively drastic local changes among pixels of potential example, in other words, local input difference $x_d{}^0$ of potential example and natural example (also refers to local perturbations) show its great variance. These perturbations are convolved on numerous kernels. As shown in Figure 4, suppose convolution kernels with the same mean but greater variance, these locally-disordered perturbations are more likely to yield large enough absolute response differences $|\Delta_{i,j}^1|$ in the first layer, since the non-smooth kernels tend to emphasize the relationship between different pixels. On the other hand, suppose smoother kernels with relatively small variance, these locally-disordered perturbations tend to show smaller absolute response differences since smoother kernels tend to focus on the average situation within their local receptive fields. That is, locally-disordered perturbations pass through different types of convolution kernels tend to produce different response differences, which can be accumulated in the subsequent layers.

**Local responses vs. locally-consistent perturbations.** Since adversarial perturbations generated by adversarially robust models tend to be locally-consistent, we count that such perturbations are more destructive to smoother kernels, comparing with locally-disordered perturbations in Figure 4. Locally-consistent perturbations $x_d{}^0$ reaching the perturbation bound tend to produce larger absolute response differences in the first layer since they tend to show a larger absolute average, yet locally-disordered perturbations show smaller response differences due to a smaller absolute average.

Both different local perturbations have different impacts on the local response differences of potential examples and natural examples, implying the fragile trend of models and the transferability of adversarial examples.

### 4.3 EMPIRICAL UNDERSTANDING OF LOCAL RESPONSES

Motivated by adversarially-trained models that tend to show smoother kernels (Wang et al., 2020a), we first provide a further understanding of local responses and then give empirical understandings.

**Local responses vs. model smoothness.** Given a non-smooth convolution kernel $m^{l+1}$ with great variance, it is more likely to cause huge impact on an absolute response difference $|\Delta_{i,j}^{l+1}|$ within its local receptive field when fixing $x_d{}^l$. Considering numerous non-smooth kernels and various local features are combined in the same layer, it is quite easy to yield large enough response differences. More complicated is the subsequent layers act on the current local response differences and get intricate accumulation effects. That is, for a standard model with non-smooth kernels, the model itself tends to amplify the local response differences between potential examples and natural examples. On the other hand, since an adversarially-trained model show smoother kernels, the model has a tendency to weaken the local response differences and narrow the final responses of potential examples and natural examples, indicating a trade-off between model robustness and accuracy.

**Setup.** We get both standard and adversarially-trained models in Section 3. Considering complex functional layers, including linear and nonlinear ones in DNNs, based on Assumption 1, we use the maximum and the total absolute differences of local responses in some layers to show their effects. Taking ResNet-18 as an example, we investigate some layers including the first convolutional layer as Conv 1, the feature maps before layer 1 as Layer 0, and Layer 1 to Layer 4 respectively[1].

To verify above analysis, we first compare the local response differences of potential examples and natural examples between standard and robust models. As Table 1 shows, the standard model exhibits significantly larger differences in local responses layer by layer, especially from the perspective of total differences yielded by per pairs. These large enough response differences accumulate, making the model's cognition of potential examples far away from their natural ones and ultimately leading the model to express a high vulnerability. On the other hand, the adversarially-trained model significantly shortens local response differences in corresponding layers, yet still some differences exist. In other words, how to further reduce the local response differences is a perspective of approaching model robustness to performance. We also note that, due to the complex effect of nonlinear layers (e.g., BatchNorm, ReLu), the maximum absolute difference of local responses does not strictly increase monotonically but increases in trend.

---

[1]Feature map size of Conv 1, Layer 0 and Layer 1: $64 \times 32 \times 32$, Layer 2: $128 \times 16 \times 16$, Layer 3: $256 \times 8 \times 8$, and Layer 4: $512 \times 4 \times 4$.

Table 1: The absolute difference of local responses (per image) between test set images and their potentially adversarial examples on different layers. Left/Right denote standard/robust model.

| Model | Conv 1 | Layer 0 | Layer 1 | Layer 2 | Layer 3 | Layer 4 |
|-------|--------|---------|---------|---------|---------|---------|
| Max | **0.12**/0.05 | **0.30**/0.15 | **1.92**/0.52 | **1.46**/0.36 | **1.45**/0.41 | **3.72**/1.02 |
| Total | 370/**556** | 443/**489** | **4276**/1246 | **2093**/430 | **581**/164 | **2624**/526 |

**Local responses vs. destructive effect.** Naturally, we wonder whether a clear difference between potential examples of successful and failed attacks on a certain model exists. We first select the natural examples that are classified correctly, and count their potential examples of successful and failed attacks. As Table 2 shows[1], for the standard model, the successful ones show similar differences with the failed ones in the front layers, but greater differences especially in Layer 4 close to the outputs, leading to finally destructive effects. However, for the adversarially-trained model in Table 4 (Appendix C), similar differences even in Layer 4 may be due to the smoother kernels and locally smoother perturbations, leading to the final responses of both close to natural examples'.

Table 2: Left/Right denote successfully-attacked/failed adversarial examples on standard model.

| Model | Conv 1 | Layer 0 | Layer 1 | Layer 2 | Layer 3 | Layer 4 |
|-------|--------|---------|---------|---------|---------|---------|
| Max | **0.16**/0.15 | **0.379**/0.378 | 2.50/**2.51** | **1.38**/1.37 | **1.16**/1.08 | **2.11**/1.38 |
| Total | **558**/554 | **676**/669 | 5955/**6001** | **2363**/2339 | **522**/494 | **1481**/757 |

Table 3: Left/Right denote original/transferred adversarial examples on adversarially-trained model.

| Model | Conv 1 | Layer 0 | Layer 1 | Layer 2 | Layer 3 | Layer 4 |
|-------|--------|---------|---------|---------|---------|---------|
| Max | **0.05**/0.04 | **0.15**/0.12 | **0.52**/0.44 | **0.36**/0.14 | **0.41**/0.08 | **1.02**/0.10 |
| Total | **556**/321 | **489**/302 | **1246**/767 | **430**/138 | **164**/38 | **526**/57 |

**Local responses vs. transferability.** Different local perturbations related to models imply the difficulty of adversarial examples' transferability, then we further explore whether the transferability can be understood from the locally intermediate response perspective. To verify the analysis of Section 4.2, we exchange the adversaries obtained from the standard and robust model. Table 3 shows, locally-disordered perturbations, combined with a smoother adversarially-trained model, exhibit fairly small local response differences layer by layer, making the model's cognition close enough to the natural examples and leading to 83.6% robustness close enough to 84.9% model performance. Similar situations occur when locally-consistent perturbations combined with a non-smooth standard model in Table 5 (Appendix C). These results indicate that the searched perturbations related to models tend to amplify response differences as much as possible to enlarge the model's cognition of potential examples and natural examples, and the weak transferability of adversarial examples may be due to transferred perturbations that are difficult to enlarge the model's cognition.

### 4.4 SMOOTHER KERNELS: ALLEVIATE LOCAL RESPONSE DIFFERENCES

To further exhibit the effect of shortening local response differences, we simply show smoother adversarially robust models can alleviate local response differences and then improve their robustness. As shown in Figure 5, adversarially-trained models with different smoothness (i.e., larger weight decay parameters show the smaller magnitude of the kernels (Loshchilov & Hutter, 2019)) show different local response differences. Figures 5(b)-5(g) for the maximum differences and Figure 9 for the total differences (Appendix C) indicate that smoother kernels slightly weaken the local response differences between the potential examples and natural examples layer by layer, and finally tend to narrow the robustness and performance in Figure 5(h). This to some extent explains adversarially-trained models are more sensitive to weight decay (Pang et al., 2021). On the other hand, we find that the increase of weight decay is quite difficult to further reduce the magnitude of parameters and the local response differences in each layer, which may be one of the reasons for the current bottleneck in the robustness of the adversarial training. Besides, the increase of weight decay can effectively weaken the robust overfitting (Rice et al., 2020) in Figure 5(h) and Figure 9(g).

---

[1]Note that under the PGD-20 attack, the standard model hardly yields examples of failed attacks, then we use the FGSM attack to illustrate the problem.

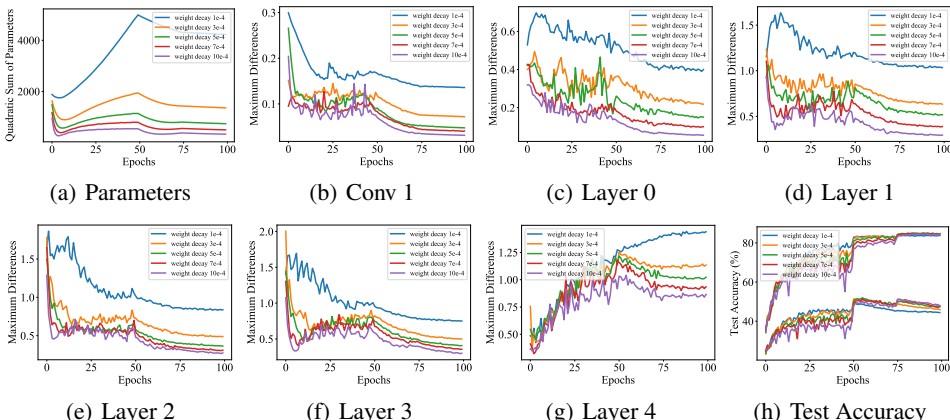

Figure 5: The maximum local response differences on robust models with different smoothness. (a) shows different model smoothness affected by weight decay, (b)-(g) denote the impact of weight decay on the local response differences, and (h) shows the model robustness and performance.

## 4.5 DISCUSSION: LOCAL RESPONSES AND MODEL ROBUSTNESS

The above suggests that the model robustness is related to the model itself and the property of potential examples, while they are combined through the local responses. Due to enough differences in local responses, some potential examples are not regarded as similar to natural examples (or their predictions are different from the natural ones with correct predictions), so they eventually show their destructive effects as adversarial examples. That is, if a model tends to weaken the local response differences of potential and natural examples, then the model exhibits great robustness as the final responses of the potential examples are more likely to be close to the natural ones.

On the other hand, though small differences of local responses make the network more inclined to treat both examples as the same categories, but the existing differences, especially the intricate differences after multi-layer accumulation emphasize the difficulty of approaching model robustness to performance, and *demonstrate that DNN models are naturally fragile*. Essentially, these response differences come from *whether a non-zero convolution kernel acts on all legal perturbations in boundary B to obtain differences that are all zeros or sufficiently small without further accumulation*. In other words, it tells the model robustness is that, given legal potential examples from any attacks, the accumulated response differences can be alleviated or removed. For instance, feature denosing (Xie et al., 2019) and activation suppressing (Bai et al., 2021) can be viewed as shortening local response differences of both examples to improve the model robustness.

Besides, shortening local response differences does not directly mean an improvement of the model's robustness, in fact, it expresses the closeness of the model's cognition on both examples. That is, a poorly-trained model may also treat both as relatively close, but give a bad model performance. To further improve the model robustness, a more realistic idea is to find proper parameters (whether theoretical parameters to minimize the local response differences exist) or a new method (nonlinear layers, loss functions, model structures) to shorten the response differences as much as possible, while not overly weaken the classification performance of the model.

## 5 CONCLUSION

In this paper, we investigate the local properties of adversarial examples generated by different models and rethink the vulnerability of DNNs from a novel locally intermediate response perspective. We find that the high-frequency components of adversarial examples tend to mislead standard DNNs, but have little impact on adversarially-trained models. Furthermore, locally-disordered perturbations are shown on standard models, but locally-consistent perturbations on adversarially-trained models. Both explorations emphasize the local perspective and the potential relationship between models and adversarial examples, then we explore how different local perturbations affect the models. We demonstrate DNN models are naturally fragile at least for large enough local response differences between potentially adversarial examples and natural examples, and empirically show smoother adversarially-trained models can alleviate local response differences to improve robustness.

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

## A  THE DESTRUCTIVENESS OF ADVERSARIAL EXAMPLES ON FREQUENCY DOMAIN

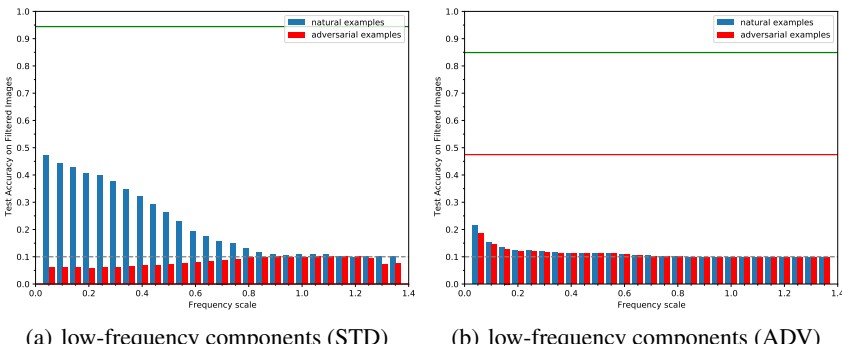

(a) low-frequency components (STD)  (b) low-frequency components (ADV)

Figure 6: The destructiveness of only high-frequency components from natural and adversarial examples on both standard (STD) and adversarially-trained (ADV) models. Shown above are well-trained models tested with images through high-pass filter.

Here, we further investigate the contribution of only high-frequency components to the destructiveness of both examples. We get both standard and adversarially-trained models as above. Figure 6 illustrates the trend of model performance and robustness on the test set with the low-frequency components decreased (the increase of the filtering scale denotes that the less low-frequency components are added to the filtered images). As the increase of filtering scale, only natural examples on standard models keep certain classification performance and then decrease to reach 10% (the accuracy of random classification). That is, the high-frequency components of natural examples to some extent can promote classification. On the other hand, the high-frequency components of adversarial examples on standard models show almost no promotion but destructive effect, and finally reach 10% as well. For robust models, the high-frequency components of both examples show similar but little performance.

## B  THE LOCAL PROPERTIES OF ADVERSARIAL EXAMPLES ON SPATIAL DOMAIN

We further explore the local properties of adversarial examples generated by the FGSM attack. For adversarially-trained models, compared with the PGD attack in Figure 3, the adversarial examples generated by the FGSM attack in Figure 7 show similar perturbations and similar model robustness (51.9% robustness for PGD-20 and 57.4% robustness for FGSM), yet the latter produces less detailed perburbations. For standard models, though both attacks show locally-disordered perturbations, the latter exhibits less disordered perturbations since less perturbation values can be achieved, i.e., +8/255, 0, and -8/255.

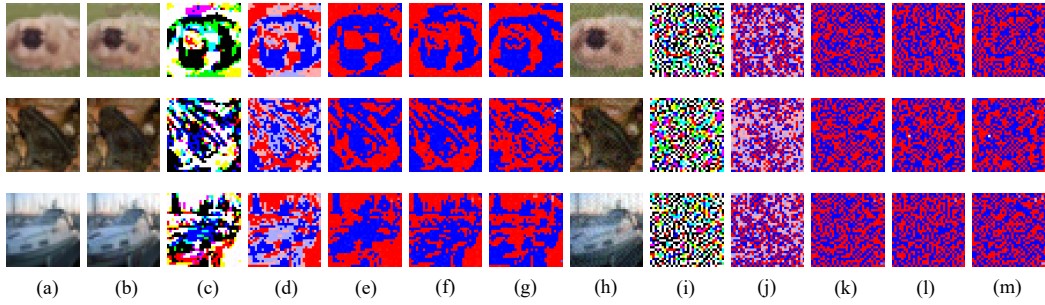

(a) (b) (c) (d) (e) (f) (g) (h) (i) (j) (k) (l) (m)

Figure 7: Visualisation of adversarial perturbations (FGSM) generated by Left: adversarially-trained (ADV) model and Right: standard (STD) model.

Similar to adversarial examples of successful attacks in Figure 3, the failed attacks in Figure 8 show locally-consistent perturbations related to image shapes on the adversarially-trained models as well. The difference is that, these examples show less misleading local perturbations since the shape of their perturbations is closer to the shape of the original image.

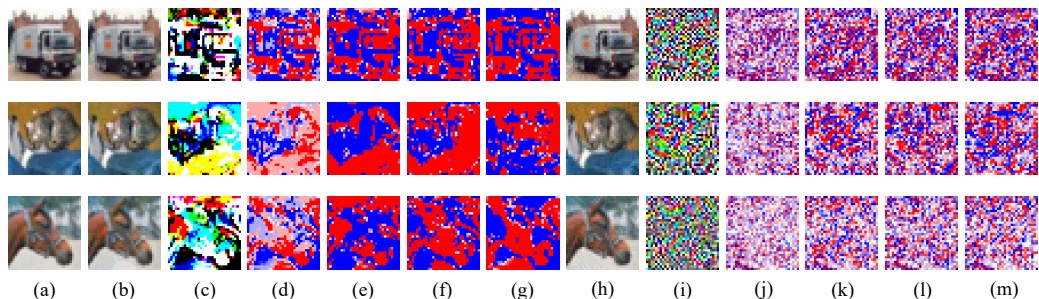

|     |     |     |     |     |     |     |     |     |     |     |     |     |
| (a) | (b) | (c) | (d) | (e) | (f) | (g) | (h) | (i) | (j) | (k) | (l) | (m) |

Figure 8: Visualisation of adversarial perturbations (PGD) generated by Left: adversarially-trained (ADV) model and Right: standard (STD) model. Shown above are all unsuccessfully attacked on ADV model.

## C   LOCAL RESPONSES

As mentioned in Section 4.3, we report destructive effects on the adversarially-trained model in Table 4. We first select the natural examples that are classified correctly, and count their potential examples of successful and failed attacks. Similar to the standard model, the successful ones show similar differences with the failed ones in the front layers, but greater differences in Layer 4 close to the outputs. The difference is that, due to the smoother kernels and locally smoother perturbations, the robust model exhibits a closer local response differences in Layer 4 between the successful ones and the failed ones. Besides, compared with the PGD attack, the adversarial examples generated by the FGSM attack show similar differences in the front layers, but less local response differences in Layer 4, leading to higher model robustness.

Table 4: The absolute difference of local responses (per image) between test set images and their potentially adversarial examples on different layers. Left/Right denote successfully-attacked/failed adversarial examples on adversarially-trained model respectively.

| Model | Conv 1 | Layer 0 | Layer 1 | Layer 2 | Layer 3 | Layer 4 |
|---|---|---|---|---|---|---|
| Max (PGD-20) | 0.048/0.047 | 0.15/0.15 | 0.51/0.53 | 0.36/0.37 | 0.40/0.42 | **1.10**/0.99 |
| Total (PGD-20) | 557/555 | 488/490 | 1264/1242 | 445/420 | 169/160 | **578**/491 |
| Max (FGSM) | 0.048/0.047 | 0.16/0.15 | 0.53/0.54 | 0.35/0.36 | 0.37/0.38 | **0.90**/0.86 |
| Total (FGSM) | 580/576 | 510/511 | 1292/1272 | 436/414 | 161/153 | **478**/428 |

We further report the transferability of adversarial examples on the standard model, as shown in Table 5. We get the potentially adversarial examples obtained from the robust model to attack the standard model. That is, locally-consistent perturbations, combined with a non-smooth standard model, exhibit small local response differences layer by layer, making the model's cognition close enough to the natural examples and leading to 78.4% robustness close to 94.4% model performance.

Table 5: The absolute difference of local responses (per image) between test set images and their potentially adversarial versions on different layers of ResNet-18. Left/Right denote original/transferred adversarial examples on standard model respectively.

| Model | Conv 1 | Layer 0 | Layer 1 | Layer 2 | Layer 3 | Layer 4 |
|---|---|---|---|---|---|---|
| Max | 0.12/**0.15** | **0.30**/0.29 | **1.92**/0.85 | **1.46**/0.88 | **1.45**/0.67 | **3.72**/0.88 |
| Total | 370/**532** | 443/**509** | **4276**/2639 | **2093**/1157 | **581**/280 | **2624**/525 |

As mentioned in Section 4.4, Figure 9 is a supplement to Figure 5 from the total absolute local response differences perspective, which indicates that smaller local response differences tend to approach model robustness to performance as well.

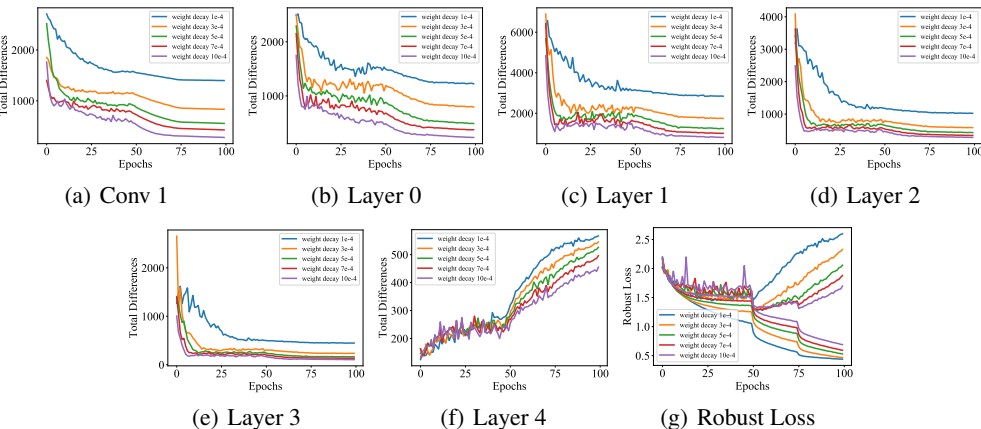

Figure 9: The total absolute local response differences (per image) of some layers on adversarially-trained models with different smoothness. Among them, (a)-(f) denote the influence of different weight decay parameters on the local response differences in each layer, and (g) shows the robust loss of training set and test set.

