# OpenReview forum: "Dissecting Local Properties of Adversarial Examples"
_ICLR.cc/2022/Conference — ICLR 2022 Submitted_

### Official Review · Reviewer_CGeY · 2021-10-26

**Correctness:** 2
**Technical Novelty And Significance:** 1
**Empirical Novelty And Significance:** 1
**Recommendation:** 1
**Confidence:** 4

**Main Review:**

The empirical findings of the paper seem to have already been known broadly. In Section 3.1, the paper shows the difference between vulnerabilities of standard training and adversarial training in the frequency domain. However, similar analyses have already been shown in (Yin et al., 2019, Wang et al., 2020a, Tsuzuku and Sato 2019). Similarly, the difference of vulnerabilities in the spatial domain has been shown in (Tsipras et al. 2019). Thus, the contributions in Section 3 are weak.

Section 4 discusses the relation between local responses and locally-disordered perturbations, locally consistent perturbation, and smoothness. However, since several terms are not well defined mathematically, the claims in this section might be ambiguous and it is difficult to obtain useful insights. For example, what is the rigorous difference between locally-disordered perturbations are locally-consistent perturbations? In addition, I think the experimental setup is not very sophisticated to answer the research question.

To obtain more useful insights, it might be helpful to narrow down the research questions and experimental setup and to develop new analyses based on the previous studies.

(Tsuzuku and Sato 2019) "On the structural sensitivity of deep convolutional networks to the directions of Fourier basis functions." CVPR 2019.

**Summary Of The Paper:**

This paper investigates the properties of adversarial examples from frequency and spatial perspectives and claims that standard models are vulnerable to high-frequency perturbations and adversarially robust models are vulnerable to locally consistent perturbations. In addition, this paper investigates the relation between local intermediate response and adversarial robustness.

**Summary Of The Review:**

- The experimental results and claims are not novel while this paper does not provide theoretical results and new algorithms. .
- The paper's assumptions are not well defined and its claims and limitations are not clear.

---

### Official Review · Reviewer_JTZC · 2021-11-01

**Correctness:** 3
**Technical Novelty And Significance:** 2
**Empirical Novelty And Significance:** 2
**Recommendation:** 3
**Confidence:** 4

**Main Review:**

**Strength:**

The paper provides interesting experiments that can potentially contribute to the understanding of the community about what properties in adversarial examples enables them to sabotage DNNs, and how adversarial-training counteracts these properties. It also suggests interesting connections between the concept of local response differences and previously observed qualities of adversarial examples.

**Concerns and Questions:**

1- A major concern with the paper is that it does not read well, contains many confusing statements, and as such I think it needs a major editorial fix.

2- There are also several ad hoc statements and arguments that must either be carefully justified mathematically or empirically or be properly referenced (for example this statement from section 4.2: “On the other hand, suppose smoother kernels with relatively small variance, these locally-disordered perturbations tend to show smaller absolute response differences since smoother kernels tend to focus on the average situation within their local receptive fields” or from section 3.4: “That is, for a standard model with non-smooth kernels, the model itself tends to amplify the local response differences between potential examples and natural examples” or from section 4.5: “On the other hand, though small differences of local responses make the network more inclined to treat both examples as the same categories, but the existing differences, especially the intricate differences after multi-layer accumulation emphasize the difficulty of approaching model robustness to performance, and demonstrate that DNN models are naturally fragile”

3- The paper does not seem to add much more over the known results from prior works, for example Tsipras et al. (2019) has showed the difference between adversarial examples of adversarially-trained vs. standard DNNs, and the presence of spatially locally consistent patterns in the former, and high frequency patterns in the latter (Figure 3 and 2 of that paper); and Wang et al. (2020a) show the smoothness of the learnt kernel in adversarially-trained DNNs. A more detailed explanation of how this paper differs from these prior works, and what aspects of this work are novel and cannot be readily inferred from the prior works, would really help highlight the importance and relevance of this work.

4- Are the results reported in Table 1,2,3 averaged? Over how many samples? What is the corresponding variance for each value?

5- In section 4.4, how is the following statement inferred from Figure 5h: “and finally tend to narrow the robustness and performance in Figure 5(h)”? I cannot see this narrowness. (also the legends in Figure 5(h) must differentiate between plots with the same color).

**Summary Of The Paper:**

This paper provides a set of empirical studies of the spectral and spatial properties of adversarial examples of deep neural nets (DNNs) classifiers. The studies illustrate that standard DNNs are much more sensitive to high frequency components of adversarial examples compared to adversarially-trained DNNs, and also that the adversarial examples corresponding to the latter exhibit more local consistency in the spatial domain. The paper then connects the effectiveness of an adversarial example to the concept of inducing larger differences in the local response of DNN layers compared to the corresponding natural example, denoted local response differences, and justifies this connection by empirically showing that: adversarially-trained DNNs have smaller local response differences on adversarial examples compared to standard DNNs; the smoother the DNN the better its robustness; successful attacks (adversarial examples) induce larger local response differences; and finally that adversarial examples transferred from other models, hence weaker examples, also induce smaller local response differences.

**Summary Of The Review:**

As the paper does not read well, and also considering that its novelty compared to prior works is unclear to me, it might not be relevant and valuable to the community in its current form, and as such I cannot recommend accepting it. That said, I think the paper does contain some good experiments that, if explained with more clarity and better placed with respect to what already exists in the prior work, can be helpful to the community.

---

### Official Review · Reviewer_Rh4T · 2021-11-01

**Correctness:** 3
**Technical Novelty And Significance:** 2
**Empirical Novelty And Significance:** 2
**Recommendation:** 3
**Confidence:** 3

**Main Review:**

Strength:
+ This paper studies an interesting problem: the connection between adversarial perturbations and local properties of adversarial examples (how perturbations associate to image shape).

Weakness:
- The observation and evaluation of this paper lack novelty and are not so surprising to me. For example, adversarial examples containing high-frequency perturbations is also studied in [1]. Adv-trained models are more robust than naturally trained ones, so they are supposed to be more robust to these high-frequency components in adversarial perturbations.

- The explanation of the relationship between frequency and spatial domain is not quite clear to me. Why do high-frequency components indicate images have relatively drastic local changes? How are the high-frequency components related to local changes in adversarial examples?

- The paper argues that the proposed smoother adversarially-trained models can achieve better robustness, but there is no comparison to the SOTA adversarial training work to show the improvement in the robustness. A quick comparison could be made with some works listed in https://robustbench.github.io/.

- The terminology, like "local changes" "locally consistent", has a high frequency in this paper, but I didn't find the definition or the explanation for what 'local' means. This makes the paper a bit vague to understand. Besides, some other words like "fragile trend" are also vague.

[1] Yin, Dong, et al. "A fourier perspective on model robustness in computer vision." arXiv preprint arXiv:1906.08988 (2019).

**Summary Of The Paper:**

This paper studies the properties of adversarial examples from a spatial and frequency perspective and shows that naturally trained models are more vulnerable to high-frequency components in adversarial examples. Perturbations for naturally trained models are disordered, but perturbations for adv-trained models are image shape related. Based on these observations, the authors find that smoother adversarially-trained can achieve better robustness.

**Summary Of The Review:**

The results presented by the paper are not so surprising and lack some novelty. The evaluation also needs additional comparison experiments to better support the conclusion that smoother adversarially-trained models enhance robustness.

---

### Decision · Program_Chairs · 2022-01-20

**Decision:**

Reject

**Comment:**

This paper explores geometric properties of image perturbations (e.g. frequency content and local consistency) and their impact on the adversarial response of networks.  The reviewers feel that the paper is at times unclear about the meaning of terminology (e.g. “local consistency”) that is not clearly defined.  Also, while the reviewers acknowledge that the paper contains a number of interesting ideas, it is not always clear how the paper’s discussions and contributions differ from existing papers (e.g. Dong 2019, Yin et al., 2019, Wang et al., 2020a, Tsuzuku and Sato 2019) that also discuss the frequency content and smoothness properties of adversarial perturbations.